# A Cluster-Driven Adaptive Training Approach for Federated Learning

**DOI:** 10.3390/s22187061

**Published:** 2022-09-18

**Authors:** Younghwan Jeong, Taeyoon Kim

**Affiliations:** 1Department of Computer Engineering, Dankook University, Yongin-si 16890, Gyeonggi-do, Korea; 2Department of Mobile System Engineering, Dankook University, Yongin-si 16890, Gyeonggi-do, Korea

**Keywords:** adaptive training, clustering, federated learning, non-IID, proportional fairness, straggler

## Abstract

Federated learning (FL) is a promising collaborative learning approach in edge computing, reducing communication costs and addressing the data privacy concerns of traditional cloud-based training. Owing to this, diverse studies have been conducted to distribute FL into industry. However, there still remain the practical issues of FL to be solved (e.g., handling non-IID data and stragglers) for an actual implementation of FL. To address these issues, in this paper, we propose a cluster-driven adaptive training approach (CATA-Fed) to enhance the performance of FL training in a practical environment. CATA-Fed employs adaptive training during the local model updates to enhance the efficiency of training, reducing the waste of time and resources due to the presence of the stragglers and also provides a straggler mitigating scheme, which can reduce the workload of straggling clients. In addition to this, CATA-Fed clusters the clients considering the data size and selects the training participants within a cluster to reduce the magnitude differences of local gradients collected in the global model update under a statistical heterogeneous condition (e.g., non-IID data). During this client selection process, a proportional fair scheduling is employed for securing the data diversity as well as balancing the load of clients. We conduct extensive experiments using three benchmark datasets (MNIST, Fashion-MNIST, and CIFAR-10), and the results show that CATA-Fed outperforms the previous FL schemes (FedAVG, FedProx, and TiFL) with regard to the training speed and test accuracy under the diverse FL conditions.

## 1. Introduction

Federated learning (FL) is a novel decentralized learning algorithm that trains the machine learning model with locally distributed data stored on multiple edge devices (client) such as smartphones, tablets, IoT devices, and sensors. FL is fundamentally different from the legacy distributed learning of the data center. In distributed learning, data are collected centrally and then trained with distributed but dedicated multiple computing resources [1,2]. On the other hand, in FL, arbitrary edge devices exchange their trainable local models with the central server over the network and train at local through their private data and resources without the exchanging of data [3]. This characteristic of FL has advantages which not only improve the privacy of sensitive data for the client, but also reduce data communication costs. With this advantage, various big-tech companies, such as Apple, Google, and IBM, continue to research the implementation of FL to their businesses. However, most of the current FL algorithms assume the learning environment to be ideal, and the issues of implementing FL in practical environments still remain. The two most important challenges in terms of practicality are the presence of straggler and non-independent and identically distributed (non-IID) client data [4,5,6].

Most of the conventional FL algorithms have two ideal assumptions. First, in the training process, all participating clients are assumed to maintain a stable connection until the end of the training and return the training result without any problem. Second, the data of every client are IID [7,8,9,10,11,12,13,14,15]. However, in a more real environment, FL generally trains models over heterogeneous systems. In other words, all clients have different computing resources and network environments. This causes the FL system to experience unexpected situations in the training process (network disconnection, fluctuation in available computing resources, and so on). Owing to this, some clients may be slow to work or unresponsive. All of these clients are called stragglers. In some FL algorithms, the central server cannot identify straggler among the participating clients, and it delays the training process until the completion of the computing of stragglers to combine the training results of all the participants [3,4]. Obviously, the delay degrades the performance of training in terms of time and resource efficiency. For this reason, other studies suggest the partial participation schemes that drop without waiting for the stragglers after the deadline. However, simply dropping the stragglers has two disadvantages: first, missing training opportunities for unique data of the dropped stragglers; and second, wasting their computational resources. In conclusion, the existence of the stragglers in the real FL environments greatly affects not only the test accuracy of the global model but also the training speed. Therefore, a moderate counter-measure against the stragglers is necessary.

FL trains the model through the local data of clients in various environments. That is to say, all clients participating in FL collect and store data in different paths. Therefore, there can be statistical heterogeneity in the distribution and size between the local data of each client. This case is called non-IID data over participating clients. In a more practical application environment, these non-IID characteristics can appear in various ways, such as the local data being biased to some specific classes, or the local data sizes of clients being dispersed over long-tail distribution. In the training process of FL, the updated gradients computed locally at the edge are aggregated by the central server. Assuming non-IID, the local models of participating clients are updated in different directions during this process. This slows down the convergence of the global model or causes the model to diverge, which degrades the performance of the model [8,16]. This is called gradient conflict due to the difference in direction of each gradient. The performance of a model becomes even worse when data sizes differ between the clients, which yields large differences of the gradient magnitudes. When participating clients perform updates with the same batch data, clients with large data sizes perform updates with more steps, so the difference in magnitude of the changed gradient is relatively wide compared to smaller clients. As a result, the different data sizes of participating clients may cause a bias to large clients in the weight average process and lead to unfair global updates to small clients [17]. Therefore, random selection schemes that do not consider the data size of participating clients can negatively affect the efficiency and performance of global model training. Therefore, when a more practical environment is assumed, an effective client selection method that can cope with non-IID in FL is required.

To address these practical issues of implementing the FL system, we propose a cluster-driven adaptive training approach (CATA-Fed), which can efficiently respond to diverse environments surrounding FL (e.g., computing resource diversity, heterogeneous network states, and non-IID data). CATA-Fed consists of two-staged schemes. In the first stage, the central server of FL allocates a deadline of the learning time to participating clients, and the clients actively determine the maximum number of training epochs adaptively to the deadlines (i.e., adaptive local update—ALU). In addition, a new straggler mitigating scheme (SMS) is devised that manages the workload of the client by partitioning the local dataset. As the second stage, a cluster-driven fair client selection scheme (CFS) is devised for CATA-Fed, which creates clusters of clients with tentative participants for the training and performs client selection within a cluster to engage them into training with the consideration of local data size of the participating clients. In particular, the client selection of CFS in CATA-Fed employs proportional fair scheduling, taking into account the fairness of training opportunity among clients. In order to evaluate the performance of CATA-Fed, extensive experiments are conducted with three realistic FL benchmark datasets (MNIST, Fashion-MNIST, and CIFAR-10) under the more practical conditions (heterogeneous data size and distribution in each client and presence of stragglers), and the results show that CATA-Fed improves the robustness, accuracy, and training speed of FL compared to legacy FL schemes [4,18,19].

The main contributions of CATA-Fed are four-fold: (1) accelerating the local model convergence, so that can improve global model training speed and reduce communication costs through adaptive local update (ALU), (2) enhancing the generalization performance of training as well as the training speed by mitigating the workload of the stragglers (SMS), (3) alleviating the divergence of the global model and elevating the robustness of training process under statistical heterogeneous conditions by cluster driven fair client selection (CFS), and (4) securing the data diversity through proportional fair client selection, which reduces the bias of the global model and balances the loads of clients at the same time.

The rest of this paper consists of the followings. Section 2 summarizes existing studies related to conventional FL. Section 3 describes the system model of CATA-Fed. In Section 4, algorithms for Stage1 and Stage2 are described and formulated. Section 5 derives and explains the experimental results for the benchmark data, and Section 6 concludes.

## 2. Related Work

FL is a distributed learning method proposed by Brendan et al. [4] that trains a global model while protecting the privacy of data of various clients. Due to these advantages, FL has the potential to be applied to various fields, such as medical care, transportation, and communication technologies and so on [20,21,22,23,24]. However, unlike the centralized model training in a single system, FL as the distributed model training has various issues according to the decentralized system architecture. One of the critical challenges on the performances of distributed model training is handling (1) system heterogeneity and (2) statistical heterogeneity [5].

Many studies so far have focused on extending FL to non-IID data from various clients. Yang et al. [25] theoretically analyzed the convergence boundary of FL based on gradient descent and propose a new convergence boundary that integrates the convergence boundary of non-IID data distribution. Sattler et al. [26] extended the existing compression technique of gradient sparsity through sparse ternary compression (STC), increasing communication efficiency, and achieving optimization in a learning environment with limited bandwidth. The authors revealed the limitations of the IID assumption on the client data of the existing FL approach. Karimireddy et al. [27] figured out the slowing of the convergence speed of non-convex functions due to non-IID, and proposed stochastic controlled averaging for on-device FL(SCAFFOLD), which can alleviate client drift and utilize similarities between participants to reduce the number of communication rounds required. Wang et al. [28] claimed the possibility of global model divergence because of the biased update by the random selection of non-IID participants in the server. To cope with this, they proposed a control framework that intelligently selects a client in order to cancel the bias caused by non-IID and increase the convergence speed. Agrawal et al. [29] proposed a CFL method for clustering clients through genetic optimization based on the hyperparameters of local model training and analyzed convergence in a non-IID environment. However, these studies did not deal with straggler issues, such as delay or the disconnection of participating clients under the condition of a heterogeneous system.

Meanwhile, various attempts have been made to assemble and optimize various clients in FL. Reisizadeh et al. [30] proposed a straggler-resilient FL, which adaptively selects participating clients assuming system heterogeneity. The proposed scheme extends the system runtime according to the communication environment, considering the calculation speed between participating clients, and integrates the statistical characteristics of clients. To address the problem of stragglers due to the system heterogeneity, Tao et al. [31] proposed the methodology to control the rate of stragglers between workers and select devices through distributed redundant n-Cayley trees. Chen et al. [32] proposed synchronous optimization through a backup worker. This backup worker avoids asynchronous noise and mitigates the influence of the straggler. Li et al. [18] proposed FedProx, which aggregates partial works of the local model on the server considering the system heterogeneity and integrates partial updates through proximal terms. Chai et al. [33] proposed FedAT, which configures tiers among clients with similar system response speed. The scheme trains tiers synchronously, and aggregates training results asynchronously, alleviating the reliance of the server on the straggler. Although these studies partially discussed non-IID issues, they do not consider fairness in the global model update and do not address the problem of model bias caused by the difference of sizes in local datasets.

On the other hand, in some studies, the comprehensive approaches are made considering the both system heterogeneity and statistical heterogeneity in FL. Li et al. [34] proposed a hybrid FL (HFL) that asynchronously aggregates stragglers, assuming system heterogeneity of various participants. This method is an extension of the existing synchronous method to analyze convergence in non-convex optimization problems by merging different delayed gradients through adaptive delayed stochastic gradient descent (AD-SGD). John et al. [35] proposed FedBuff, a buffered asynchronous aggregation method to extend FL to secure aggregation. This approach achieves a non-convex optimization by configuring the size of the buffer to be variable and staleness scaling to constrain the ergodic norm-squared of the gradient. Chai et al. [19] proposed a tier-based federated learning system (TiFL) to schedule client selection through tiers. Tiers are configured differentially according to the system response speed, and credit is granted to consider data heterogeneity for proper tier selection. In a similar way, Lai et al. [36] proposed Oort for an effective client selection that provides the greatest utility for model convergence and fast training in a non-IID environment. This approach makes client selection, taking into account the utility of heterogeneous data, by introducing a pragmatic approximation of the statistical utility of the client. Xie et al. [37] proposed FedAsync for an efficient aggregation of heterogeneous clients in a non-IID environment. This approach normalizes the staleness weights and adaptively tunes the monotonically increasing or decreasing mixing parameters to control asynchronous noise and achieves non-convex optimization. Theses studies potentially perform specific client-dependent training implementing modified weights. However, they may contain limitations in improving the generalization performance.

The key features of the above mentioned schemes are summarized in Table 1. In this work, we conduct research for an approach that can cope with the above-mentioned various heterogeneity conditions in order to improve the performance of the global model training.

## 3. System Model

Figure 1 shows the FL system architecture. This FL system consists of four processes, and a cycle of these processes is defined as a global iteration in this paper. We consider an environment for distributed training of multiple clients connected to a central server. In *t*-th global iteration, let *C* be the set of all clients connected to the network, and $S_{t}$ be the set of participating *N* clients selected by the central server. All connected clients *C* have independent local data in each device. Among them, the local data of a certain client $c_{i}\in C$ are called $D_{i}$ and expressed as $D_{i}=\left\{d_{i,1},d_{i,2},\ldots ,d_{i,j}\ldots ,d_{i,|D_{i}|}\right\}$, where $d_{i,j}$ means the *j*-th single data point of the client $c_{i}$, and $|D_{i}|$ is the size of the local data in client $c_{i}$. The central server coordinates multiple clients in the following way to obtain the optimal model *W*.

The global loss function that the central server wants to minimize in FL is as follows:(1)$$F\left(W\right)=\frac{1}{\left|D\right|}\sum_{i=1}^{\left|C\right|}\sum_{j=1}^{|D_{i}|}f\left(W,d_{i,j}\right)$$
where |D| is the total amount of local data in all of the clients connected to the network. For this, each participating client selected by the central server performs training as follows in the *t*-th global iteration. As a first process in Figure 1, the central server randomly selects the set of participating clients $S_{t}$ from *C*. The central server sends a copy of the global model in *t*-th global iteration $W_{t}$ to the selected clients $c_{i}\in S_{t}$. The client $c_{i}$ receives the transmitted global model $W_{t}$ and replaces it with the local model $w_{t}^{i}\leftarrow W_{t}$.

Then, as a second process of Figure 1, the client $c_{i}$ trains the model $w_{t}^{i}$ through local data $D_{i}$. For this, the local loss function $F_{i}\left(w_{t}^{i}\right)$ of client $c_{i}$ is defined as follows:(2)$$F_{i}\left(w_{t}^{i}\right)=F_{i}\left(w_{t}^{i},D_{i}\right)=\frac{1}{|D_{i}|}\sum_{j=1}^{|D_{i}|}f\left(w_{t}^{i};d_{i,j}\right).$$

The client $c_{i}$ aims to gradually reduce the loss function as in Equation (Equation 2) through a local update.

In this FL system, assume that the client optimizes the local model through stochastic gradient descent (SGD). Then, in the epoch *k* of SGD, the client $c_{i}$ updates the local model $w_{t,k}^{i}$ in the negative direction of the gradient of the loss function evaluated from a group of data points (mini-batch) as
(3)$$w_{t,k}^{i}\leftarrow w_{t,k}^{i}-\eta ▿F_{i}\left(w_{t,k}^{i},b\right),$$
where η is the learning rate, and $b\subset D_{i}$ is the group of data points (mini-batch) selected randomly across the entire local data. Through updating the local model with the entire data in client $c_{i}$, which is partitioned into mini-batches, the local model moves to approximate the minimum of the loss function as
(4)$$▿F_{i}\left(w_{t,k}^{i},D_{i}\right)=E\left[▿F_{i}\left(w_{t,k}^{i},b\right)\right],$$
where E[.] is an expectation function. Through this process, client $c_{i}$ completes a single local update.

Meanwhile, the central server imposes multiple times of updating the local model $w_{t}^{i}$ on each client $c_{i}$ by putting a constant *K*, and the *K* times of local updates in $w_{t}^{i}$ is given by
(5)$$w_{t,k+1}^{i}=w_{t,k}^{i}-\eta ▿F_{i}\left(w_{t,k}^{i}\right),\;\;\;k=0,1,\cdots ,K-1.$$

After that, in the third process of Figure 1, the client $c_{i}$ uploads the obtained local update result $w_{t,K}^{i}$ to the central server after *K* updates. For the convenience of expression, let the uploaded model $w_{t,K}^{i}$ from client $c_{i}\in S_{t}$ be $W_{t}^{i}$ in *t*-th global iteration.

In this approach, however, the bigger the data size of the client, the more training time is consumed. Under the condition of data size heterogeneity across clients, the smaller clients have to wait until the end of the training of the bigger clients. This limits the speed of approximation to the optimal point of the objective function of the participating clients and makes the central server take more communication rounds for training.

During an aggregation step in the fourth process, the central server updates the global model with the local update results $W_{t}^{i}$ of all participating clients, assigning weights in proportion to the data size of each clients $c_{i}\in S_{t}$ as follows:(6)$$W_{t+1}=\sum_{i=1}^{|S_{t}|}\frac{|D_{i}|}{D_{t}}W_{t}^{i},\;\;\;\;\;\mathrm{where}\;\;\sum_{i=1}^{|S_{t}|}\frac{|D_{i}|}{D_{t}}=1,$$
and where $D_{t}$ is the sum of the data size of all participating clients as expressed $D_{t}=\sum_{i=1}^{|S_{t}|}\left|D_{i}\right|$ and $|S_{t}|$ is the total number of participate clients in *t*-th global iteration. After updating the global model $W_{t+1}$ from $W_{t}$, the central server distributes $W_{t+1}$ to the next participants $S_{t+1}$ selected for the next global iteration. FL gradually approaches the optimal model *W* through repetition of these series of processes.

However, if the conflicting gradients among the randomly selected clients have large differences in their magnitudes under the condition of statistical heterogeneity (non-IID data and different dataset sizes), FL averaging gradients from the clients may not ensure fairness for the clients (i.e., uniformity of performance on global model convergence across clients). This unfair FL may suffer reduced training speed and decreased model accuracy.

The overall operations of the system model is expressed with Algorithm 1. The global model training process of the central server is represented in lines 1 to 12, and the local update process of a participating client is described in lines 15 to 24. The server randomly selects the participating clients as presented in line 5. From then, the server broadcasts the global model to the clients and trains the model in parallel (lines 6 to 9). In the local update process, the participating client trains the received model with its own local data, and then uploads the model (lines 18 to 23). After that, the server updates the global model with the weighted average of the aggregated local models (line 10).

**Algorithm 1** System model of FL**Input:** Set of connected clients *C* , *E* is the number of global iteration , *K* is the number of local epoch, η is the learning rate , *b* is the size of local mini batch , *r* is rate of client selection**Output:** Global model *W*1: **procedure**
Server(E,K)▹ Central Server execute2:   $W_{0},k\leftarrow Initialization$▹ Initialize global model, constant *K*3:   **for** global iteration t≤E **do**4:    N←max(|C|×r,1)5:    $S_{t}\leftarrow$ ClientSelection (*C*, *N*)▹ Select client for train6:    **for** each client $c_{x}\in S_{t}$ in parallel **do**7:     Broadcast $W_{t},K$ to client $c_{x}$8:     $W_{t}^{x}\leftarrow$ ClientUpdate $\left(c_{x},K,W_{t}\right)$▹ aggregate model9:    **end for**10:    Update global model $W_{t+1}\leftarrow \sum_{x=1}^{|S_{t}|}\frac{|D_{x}|}{D_{t}}W_{t}^{x}$11   **end for**12: **end procedure**13: 14: 15: **procedure** clientupdate($c_{x},K,W_{t}$)16:   *B*← split local data $D_{x}$ into batches of size *b*17:   Replace local model $w_{t}^{x}$←$W_{t}$18:   **for** local epoch *e*≤*K* **do**19:    **for** batch data *b*∈*B* **do**20:     $w_{t,e}^{x}\leftarrow w_{t,e}^{x}-\eta ▿F_{x}\left(w_{t,e}^{x},b\right)$▹ mini-batch SGD21:    **end for**22:   **end for**23:   Upload local update result $W_{t}^{x}\leftarrow w_{t,K}^{x}$24: **end procedure**

## 4. CATA-Fed

In this section, we propose a two-stage cluster-driven adaptive training approach for federated learning (CATA-Fed). The major interest of the first stage of CATA-Fed is alleviating the impact of the straggling client. Therefore, the first stage of CATA-Fed presents the training speed accelerating scheme under the environment of the heterogeneity across clients in terms of the local model updating time (this metric can be impacted by the performance related factors such as the data-size, computing power of each client). In addition to this, the straggler mitigating scheme is proposed in the first stage of CATA-Fed, which can enhance the generalization performance of global model as well as the training speed. The second stage of CATA-Fed is focused on addressing the non-IID issue. The bias of the global update under the condition of statistical heterogeneity worsens as the difference of the gradient magnitudes of clients increases. Therefore, a new cluster-driven client selection scheme is proposed in the second stage of CATA-Fed, which can reduce the differences of the data size among the participating clients in a given global iteration. Moreover, the client selection scheme defines the proportional fair scheduling of the clients to achieve the data diversity as well as the load balancing among connected clients.

### 4.1. Stage 1: Proposed Approaches for Overcoming Stragglers

In order to address the limitations of the fixed number of local update (mentioned in Section 3), in the first stage of CATA-Fed, instead of allocating a fixed constant *K* for the local update, the central server distributes deadline *T*. Then, each participating client performs an adaptive local update (ALU) in which the client makes an adaptive decision on the maximum number of its local updates internally. This process accelerates the convergence of the global model by increasing the speed of convergence in the local model of each participating client [38].

Meanwhile, in a deadline mode of FedAVG, the server drops clients that have not yet completed the fixed number of local updates before a given deadline. In this mode, the bigger clients has a higher probability of being dropped because they may take more time to process their data. If the adaptive number of local updates is applied, preventing the drop of bigger clients, the loss of computing resources can be reduced, and the convergence of the global model can be accelerated. In addition, the model can be trained with unique data of clients that may have been dropped in the legacy deadline mode, so the generalization performance can be improved.

When FedAVG finds a straggler, it simply drops the straggling client without considering any countermeasures about the problem. On the other hand, in CATA-Fed, a straggler mitigating scheme (SMS) is proposed to handle this issue. The key idea of SMS is to split the data of the straggling clients so that local training can be completed within the interval of a global iteration. Training on the partitioned data may have a negative effect on the training efficiency. However, if data partitioning is performed properly, the global model accuracy can be improved by increasing the generalization performance compared to schemes that simply exclude the stragglers.

#### 4.1.1. Adaptive Local Update Training Scheme

Figure 2 shows how a single participate client performs adaptive local update (ALU) through deadline *T* in CATA-Fed. *N* participating clients $c_{i}\in S_{t}$ selected by the central server receive a copy of the global model $W_{t}$ and a deadline *T*. This deadline *T* is the quantity of time interval for participating clients to perform local updates in a given global iteration. During this time, participating clients try to update the local model as much as possible during local training time *T* by means of ALU.

For this end, each participating client $c_{i}\in S_{t}$ first replaces the copy of the global model $W_{t}$ with the local model $w_{t}^{i}$. Then, the client starts local training as in Equation (Equation 3) and checks the start time through the timer counter. When a local update is finished in a single epoch, the call-back function of the client can measure the time spent in one epoch between the start and the end of the local update. Let $\tau_{k}^{i}$ be the training time spent on the *k*-th local update of client $c_{i}$. This training time $\tau_{k}^{i}$ can be vary in real-time with the current computing power of each participating client. Therefore, in ALU, the expected local update time of the client $c_{i}$, $\tau_{mean}^{i}$, is calculated by averaging the values of $\tau_{k}^{i}$ as the client goes through the multiple local updates. Then, in *e*-th local update, $\tau_{mean}^{i}$ can be expressed as
(7)$$\tau_{mean}^{i}=\frac{1}{e}\sum_{k=0}^{e-1}\tau_{k}^{i}.$$

This average local update time $\tau_{mean}^{i}$ can serve as a criterion for determining whether to continue with the next local update of client $c_{i}$ or not (termination of the local model training). Therefore, after finishing the current local update, client $c_{i}$ compares $\tau_{mean}^{i}$ with the remaining time until the deadline, $\left(T-\varepsilon_{i}\right)-\sum_{k=0}^{e-1}\tau_{k}^{i}$, and continues to conduct the next local update if $\tau_{mean}^{i}\leq \left(T-\varepsilon_{i}\right)-\sum_{k=0}^{e-1}\tau_{k}^{i}$, where $\varepsilon_{i}$ is the amount of time required for uploading the local model of client $c_{i}$ which can be impacted by the communication state and model matrix size of the client. If $\left(T-\varepsilon_{i}\right)-\sum_{k=0}^{e-1}\tau_{k}^{i}<\tau_{mean}^{i}$, the client terminates the local training. Let $E^{i}$ be the maximum number of local updates of a participating client when *T* and $\varepsilon_{i}$ are given, then it can be obtained as
(8)$$E^{i}=\left\lfloor \frac{\left(T-\varepsilon_{i}\right)}{\tau_{mean}^{i}}\right\rfloor$$

Then, after performing the maximum local updates $E^{i}$ during the given deadline *T*, the updated local model of a participating client $c_{i}$ is $w_{t,E^{i}}^{i}$, which can be obtained from the following relation
(9)$$w_{t,k+1}^{i}=w_{t,k}^{i}-\eta ▿F_{i}\left(w_{t,k}^{i}\right),\;\;\;k=0,1,\cdots ,E^{i}-1.$$

Finally, $W_{t}^{i}$, the uploaded local training result from client $c_{i}$ in *t*-th global iteration can be obtained from $w_{t,E^{i}}^{i}$.

#### 4.1.2. Straggler Mitigating Scheme

After finishing the local training time, the central server collects the local update results from the participating clients to update the global model. Meanwhile, not all of the participating clients can upload valid results because there may be straggling clients among the clients. In relation to this, the clients can be categorized into three classes. The first class indicates the valid clients, those who can successfully upload valid local training results with the successful local updates within the deadline. The second and the third classes are classified as stragglers that cannot upload valid local training results. In more detail, the second class (slow straggler) is the clients that cannot complete even a single local update within the local training interval because the amount of data in them is too large or the computing power is weak. The third class (disconnected straggler) is the clients unable to upload local training results due to the loss of connection to the central server with various network problems.

Figure 3 shows the examples of the classification for the participating clients in CATA-Fed. Client 2 and client *i* are the valid clients who complete the local update at least once within the deadline time *T* and successfully upload the local update result to the central server. Meanwhile, client 1 is the slow straggler, for which a single local update is not completed before the deadline. In the case of synchronous training strategy, local model aggregation is performed at a dedicated point by the central server, and thus any stale local update results (e.g., client 1) are dropped at the server side [32]. In the case of client 3, the client is disconnected from the central server during performing the local update. As a result, the central server cannot receive any local update results from the client at a given global iteration.

In this section, we focus on the method of handling the slow stragglers (e.g., client 1 of Figure 3). In ALU, client $c_{i}\in S_{t}$ measures the time duration of an epoch for a local update to determine the number of the local updates during the local training interval. However, the client is unable to measure the epoch duration when it fails to complete a single local update. Then, the client recognizes itself as a straggler and stops the local update being performed. After that, the client performs the straggler mitigating process (i.e., SMS). At this point, the client conducts stratified sampling with information from the labels on the local data to split the entire data into two identically distributed sub-datasets. The stratified sampling is widely used in statistics to generate representative samples which represent the characteristics of the original dataset (e.g., the distribution of the data population) [39]. After the partitioning process, the client reports the size of partitioned dataset to the server.

Finally, if this client is selected as a participant again in the future, the client selects one of the partitioned datasets to perform training. For multiple selections, the partitioned datasets are rotated through round-robin-like scheduling. This allows the client to decrease the time taken to perform local updates and report training results to the server within the local training interval. Moreover, by training the representative data samples (partitioned dataset), the client can avoid bias in training caused by partitioning as much as possible. Meanwhile, the data partitioning can be performed at a linear time of running cost as O(n), which is feasible in consumer electronics of users with a low computational power [40].

As mentioned above, when the slow straggler fails to complete a local update, it switches to be a new client with a half-sized dataset. This paper refers to this process as client partitioning and refers to the new client as a sub-client. In SMS, if a client fails local training, client partitioning is performed once. If the sub-client fails to train again, client partitioning is performed again as shown in Figure 4.

Let $u_{i,t}$ be the counter of client partitioning of client $c_{i}\in C$ in a *t*-th global iteration. Then, the counter in the next global iteration can be written as
(10)$$u_{i,t+1}\overset{\Delta }{\;=}\left\{\begin{array}{cc} \; & u_{i,t}+1,\;\;\;\mathrm{if}\;\mathrm{client}\;c_{i}\in S_{t}\;\mathrm{fails}\;\mathrm{local}\;\mathrm{training},\;u_{i,t}\in \left[0,\beta_{i}\right],\\ \; & u_{i,t},\;\;\;\;\;\;\;\mathrm{otherwise}, \end{array}\right.$$
where $u_{i,t}$ is initialized to 0 when the client is connected to the network. SMS limits the maximum number of $u_{i,t}$ by placing upper bound βi=log2DiDiππ, where π is the predetermined minimum data size of the sub-dataset. This is because if the partitioning is performed too many times, the partitioned data may lose their representative property, which can hinder the generalization performance of the global model.

The number of sub-clients of client $c_{i}$ at *t*-th global iteration can be written as $2^{u_{i,t}}$. Let $D_{i,x},x\in \left[1,2^{u_{i,t}}\right]$ be the sub-dataset of client $c_{i}$. Then, $D_{i,x}$ satisfies the following conditions:(11)$$\left\{\begin{array}{cc} D_{i}={⋃}_{x=1}^{2^{u_{i,t}}}D_{i,x}, & \\ D_{i,x}\cap D_{i,y}=\emptyset , & \mathrm{where}\;x\neq y. \end{array}\right.$$

Finally, the local model update of a given client $c_{i}\in S_{t}$ can be expressed with a modified version of Equation (Equation 9), which is given by
(12)$$w_{t,k+1}^{i}=w_{t,k}^{i}-\eta ▿F_{i}\left(w_{t,k}^{i},D_{i,x}\right)\times I_{i},\;\;x=1+n_{i}\;mod\;\left(2^{u_{i,t}}\right),$$
where $n_{i}$ is the number of times client $c_{i}$ is selected by the central server, and $I_{i}$ indicates the variable set by the client as follows:(13)$$I_{i}\overset{\Delta }{\;=}\left\{\begin{array}{cc} \; & 1,\;\;\;\mathrm{if}\;\mathrm{client}\;c_{i}\in S_{t}\;\mathrm{successfully}\;\mathrm{completes}\;\mathrm{local}\;\mathrm{update}\;\mathrm{at}\;\mathrm{least}\;\mathrm{once},\\ \; & 0,\;\;\;\mathrm{otherwise}. \end{array}\right.$$

In particular, the client that was a straggler sequentially rotates the sub-clients to perform local training whenever it is selected by the server according to Equation (Equation 12). After the deadline of a given global iteration, a participating client uploads the local training results to the server which includes the updated local model $W_{t}^{i}\leftarrow w_{t,E^{i}}^{i}$ in Equation (Equation 9) and $I_{i}$. Note that $W_{t}=W_{t}^{i}$ according to Equation (Equation 12) in the case of local training failure of a slow straggler.

Meanwhile, in the aggregation step, the central server can distinguish the classes of the participating clients by taking a look at the uploaded local training results from the clients as follows
(14)$$c_{i}\left(\in S_{t}\right)\in \left\{\begin{array}{cc} \; & S_{t}^{D},\;\;\;\mathrm{if}\;W_{t}^{i}=\emptyset ,I_{i}=0,\\ \; & S_{t}^{S},\;\;\;\;\mathrm{if}\;W_{t}^{i}=W_{t},I_{i}=0,\\ \; & S_{t}^{V},\;\;\;\;\mathrm{if}\;W_{t}^{i}\neq W_{t},I_{i}=1, \end{array}\right.$$
where $S_{t}^{D}$ is the set of the disconnected clients, $S_{t}^{S}$ is the set of the slow stragglers, and $S_{t}^{V}$ is the set of the valid clients. By means of this, the server can manage a state of the connected clients as $C=C-{⋃}_{i=0}^{t}S_{i}^{D}$.

In the aggregation step, the central server should selectively collect the local update results from the valid clients. Therefore, the weight $\frac{|D_{i}|}{D_{t}}$ of each client in Equation (Equation 6) should be modified as the datasize of the client over the total quantity of datasets in the valid clients of the *t*-th global iteration. Then, the weight value of client $c_{i}\in S_{t}$ can be given by
(15)$$\psi_{i}\overset{\Delta }{\;=}\frac{|D_{i,x}|}{\sum_{j=1}^{|S_{t}|}(I_{j}\times |D_{j,x}\left|\right)}.$$

As a result, the global model update of CATA-Fed can be formulated as
(16)$$W_{t+1}=\sum_{i=1}^{|S_{t}|}\psi_{i}I_{i}W_{t}^{i}.$$

### 4.2. Stage 2: Cluster-Driven Fair Client Selection

An effective way to mitigate the gradient conflict (mentioned in Section 3) is to select clients with a similar data size and perform a weighted averaging process with them. To enable this, a cluster-driven fair client selection scheme (CFS) is proposed in the second stage of CATA-Fed. By means of CFS, CATA-Fed can perform an averaging of gradients from the clients with similar weights. Accordingly, CATA-Fed can prevent the bias to the large clients and lower the divergence probability of the global model. As a result, this accelerates the convergence of the global model and enhances the model accuracy.

However, there still remains a problem of domination from the repeatedly selected clients. Note that data are assumed to be non-IID distributed onto clients. If some clients are repeatedly selected in multiple global iterations, then the global model will inevitably be biased in the direction of the data of those clients. To address this issue, the proportional fair (PF) rule is implemented in the client selection process of CFS. It considers the fairness of the training opportunity among clients with a latency for the training of each client. From PF scheduling, CATA-Fed can improve FL to ensure data diversity during the training process over non-IID data distribution, which results in the improvement of model accuracy. Moreover, PF scheduling can balance the loads of clients.

In order to implement CFS to perform appropriate rules in CATA-Fed, we defined the scheme requirements as follows.

Requirements:The central server divides the entire connected clients into multiple clusters of the clients with similar data size.There should be no duplicate inclusion of clients across clusters.At each global iteration, the central server selects $|S_{t}|$ numbers of clients as a participating set from a chosen cluster.To allocate fair training opportunity to clients, the central server prefers to select clients that were less selected before. This also means that larger clusters containing more clients should be selected more often.To enhance generalization performance, the central server should ensure randomness in the composition of participating group as far as possible. In other words, we want to minimize the correlation of the participating groups across the entire global iteration to reduce the probability of bias.

#### 4.2.1. Client Clustering Scheme Considering Data Size

The central server divides all the connected clients into *P* clusters, according to their data size. To do this, we assume that the clients inform about the sizes of their local data to the central server when they access the network. Moreover, if there happen to be any changes in the data size of clients owing to SMS, the clients notify the changes to the server for regenerating clusters before the start of the next global iteration. In the clustering process, CFS of the central server utilizes the interquartile range (IQR) to measure the statistical dispersion of data size distribution across clients. The reason for using IQR is to limit the impact of extreme values or outliers. In a practical environment, the data size distribution of the client may have various distributions other than normal distributions. It is widely known in statistics that IQR is robust to skewed distributions.

By measuring the data sizes of clients, the server defines the lower and upper quartiles as $Q_{1}$ and $Q_{3}$, respectively, where $Q_{1}$ and $Q_{3}$ are values of the data size at 25% and 75% of the distribution. Then IQR can be calculated as $Q_{3}-Q_{1}$. With these values, the lower and the upper outlier points are determined as
(17)$$\begin{array}{c} \mathrm{LowerOutlier}(\mathrm{LO})\overset{\Delta }{\;=}Q_{1}-\delta \cdot IQR\\ \mathrm{UpperOutlier}(\mathrm{UO})\overset{\Delta }{\;=}Q_{3}+\delta \cdot IQR. \end{array}$$

Here, we apply δ=1.5 for the moderate outlier, which is widely applied in data analysis.

From this, the server defines the moderate range of data sizes by eliminating the outlier values as $R\overset{\Delta }{\;=}[LO,UO]$. After that, the server redefines the data size range to obtain a more practical range to be used in clustering, avoiding the possible negative values of *R*. Let $r_{t}$ be the set of clients with data sizes in the range *R* at *t*-th global iteration. Then $r_{t}$ is given by
(18)$$r_{t}\overset{\Delta }{\;=}\left\{c_{i}:\left|D_{i}\right|\in R,c_{i}\in C\right\},$$
where $|D_{i}|$ is the data size of the local client $c_{i}$. $R^{\prime }$, the redefined range of data sizes, can be expressed as $\left[{R_{l}}^{\prime },{R_{u}}^{\prime }\right]$, where ${R_{l}}^{\prime }=\min\left\{\left|D_{i}\right|:c_{i}\in r_{t}\right\}$ and ${R_{u}}^{\prime }=\max\left\{\left|D_{i}\right|:c_{i}\in r_{t}\right\}$. Finally, the central server defines the width of each cluster $X_{m},m\in \left[2,P-1\right],$ by dividing the interval of range $R^{\prime }$ as
(19)$$\theta \overset{\Delta }{\;=}\frac{{R_{u}}^{\prime }-{R_{l}}^{\prime }}{P}.$$

Through this, in the *t*-th iteration, the clients are clustered into cluster $X_{m}$ as follows
(20)$$X_{m}\overset{\Delta }{\;=}\left\{\begin{array}{cc} \; & \left\{c_{i}:\left|D_{i}\right|-{R_{l}}^{\prime }\leq 1\times \theta ,c_{i}\in C\right\},\;\;\;\;\;\;\;\;\left(m=1\right),\\ \; & \left\{c_{i}:\left(m-1\right)\times \theta \leq \left|D_{i}\right|-{R_{l}}^{\prime }\leq m\times \theta ,c_{i}\in C\right\},\;\;\left(2\leq m\leq P-1\right),\\ \; & \left\{c_{i}:\left(P-1\right)\times \theta \leq \left|D_{i}\right|-{R_{l}}^{\prime },c_{i}\in C\right\},\;\;\;\;\;\;\left(m=P\right). \end{array}\right.$$

Here, cluster $X_{1}$ and $X_{P}$ may have larger intervals than θ to include outlier clients.

According to Equation (Equation 20), all clients connected to the network are divided into *P* clusters as shown in Figure 5. As a next step, in the beginning of the *t*-th global iteration, the server chooses one of the clusters, and then $|S_{t}|$ clients are selected as a set of participating clients within the chosen cluster.

#### 4.2.2. Proportional Fair Client Selection

To allocate fair training opportunity to every clients, CFS of the server keeps track of the waiting time before the training of each client $\forall c_{i}\in C$, $A_{i}\left[t\right]$, where $A_{i}\left[t\right]$ is a function of global iteration. In more detail, $A_{i}\left[t\right]$ means the number of global iterations elapsed from the last selection of client $c_{i}$ to the current *t*-th global iteration. If the client $c_{i}$ is selected for training at a given global iteration, then the waiting time is initialized as 0. Thus, $A_{i}\left[t\right]$ can be expressed as
(21)$$A_{i}\left[t+1\right]\overset{\Delta }{\;=}\left(A_{i}\left[t\right]+1\right)\times \left(1-\alpha_{i,t}\right),\;\alpha_{i,t}\in \left\{0,1\right\}$$
where $\alpha_{i,t}$ is a variable indicating whether a client $c_{i}$ is selected by the central server and is defined as
(22)$$\alpha_{i,t}\overset{\Delta }{\;=}\left\{\begin{array}{cc} 1, & \mathrm{if}\;c_{i}\;\mathrm{was}\;\mathrm{selected}\;\mathrm{in}\;(t-1)\text{-}\mathrm{th}\;\mathrm{round},\\ 0, & \mathrm{otherwise}.\;\;\;\;\;\;\;\;\;\;\;\;\; \end{array}\right.$$

As shown in Figure 6, $A_{i}\left[t\right]$ of client $c_{i}$ increases at every iteration before selection, and if the client is selected, $A_{i}\left[t\right]$ is initialized to 0.

Since CFS of CATA-fed aims for the balanced learning opportunities, it can be considered that the higher the value of $A_{i}\left[t\right]$, the higher the priority of client $c_{i}$ for the selection. However, if CFS selects $|S_{t}|$ number of clients for the training participation based only on the $A_{i}\left[t\right]$ value, then there may occur a problem of fixation of participating members which does not meet the requirement of CFS (fifth item of the requirement). Moreover, CFS should establish a standard for the decision making of the selection of a cluster.

To address this, CFS introduces a method of the client grouping in clusters. At the beginning of each global iteration, the server divides the clients in the cluster into multiple groups consisting of randomly selected $|S_{t}|$ clients as shown in Figure 7. These groups are only used in a current global iteration, and the new groups are generated with random clients in the next iteration. Let $G_{x}$ be an arbitrary group belong to cluster $X_{m}$. Then, $G_{x}$ can be written as
(23)$$G_{x}\in {\left[X_{m}\right]}^{\left|S_{t}\right|},\;\mathrm{where}\;1\leq x\leq \sum_{m=1}^{P}\left\lceil \frac{|X_{m}|}{\left|S_{t}\right|}\right\rceil ,$$
and ${\left[A\right]}^{\left|k\right|}$ is the set of the *k*-element subset of *A*. Note that these groups are generated as mutual exclusive sets, and it can be given as $G_{i}\cap G_{j}=\emptyset$, where i≠j.

After that, the server calculates the priority $p_{x}$ of each group $G_{x}$ by summing $A_{i}\left[t\right]$ values of member clients as follows:(24)$$p_{x}\overset{\Delta }{\;=}\sum_{i=1}^{|S_{t}|}A_{i}\left[t\right],c_{i}\in G_{x}.$$

Then, the PF scheduler (CFS) of the central server selects a group $G_{x^{*}}$ that maximizes the following equation as
(25)$$G_{x^{*}}=\underset{x}{\arg\max}\left[\frac{p_{x}}{\sum_{k}p_{k}}\right].$$

Accordingly, this results in the selection of a cluster that contains the group $G_{x^{*}}$ in it.

### 4.3. Operational Example of CATA-Fed

This section describes the example for the overall operation of CATA-Fed. In this example, we assume that the number of clusters *P* is 3, and the central server selects four clients as the training participants in every global iteration ($|S_{t}|=4$). It is also assumed that 40 clients are connected to the network and the sizes of the datasets in the clients follow the right-tailed distribution shown in Figure 8.

Figure 8 shows an example of CFS in CATA-Fed. The server first performs clustering with the size information of local datasets collected from every connected client before starting global model training. For clustering, the server calculates a range $\left[{R_{l}}^{\prime },{R_{u}}^{\prime }\right]$ from [LO,UO] of IQR according to the Equations (Equation 17) and (Equation 18). Then, the server divides the range into three (P=3) segmented ranges. Then, the server creates three clusters corresponding to each segmented range according to Equation (Equation 20). Meanwhile, in this example, cluster $X_{3}$ has a larger interval than the others because it should include clients laying on the right tail of the distribution in the figure. However, clusters $X_{1}$ and $X_{2}$ contain more clients than $X_{3}$ because more clients are concentrated in the head and mid than in the tail of the distribution. As a result, $X_{1}$, $X_{2}$, and $X_{3}$ has a ratio of 2:2:1 as shown in the figure.

At the start of every global iteration, the server groups four random clients in each cluster to form multiple groups. In this example, eight clients in $X_{3}$ are grouped into two groups. As shown in Figure 8, the priority $p_{x}$ of the group $G_{x},x\in \left[1,10\right],$ is determined by the sum value of $A_{i}\left[t\right]$ of each client $c_{i}\in G_{x}$. The server selects the group $G_{7}^{*}$ with the highest priority $p_{7}=21$ as a participating client. After the selection, $A_{i}\left[t\right]$ of the selected clients $c_{i}\in G_{7}$ are initialized to 0, and $A_{i}\left[t\right]$ values of the unselected clients $c_{i}\in C-G_{7}$ increase by 1. The existing groups are disbanded, and new groups are formed again with randomly selected four clients in each cluster at the next global iteration.

Figure 9 shows the procedures of the global update and ALU in CATA-Fed. After the group selection, a copy of the global model and deadline are sent to the participating four clients $c_{i}\in G_{7}$ in step 2 of the figure. Then, in step 3, the clients replace the local model with the received global model. In step 4, the client performs adaptive training during the local training interval, in which each client actively determines the number of local updates comparing the remaining time to the deadline and the average epoch time for its local update. As a result, with the assumption of $|D_{1}\left|<\right|D_{2}|$ in this example, client $c_{1}$ is able to perform three local updates, whereas client $c_{2}$ only performs two.

Meanwhile, $c_{1}$ and $c_{3}$ have performed valid local updates. In this case, the indicator variable $I_{1},I_{3}=1$ is uploaded along with the updated local models. $c_{4}$ is a disconnected client and cannot upload any results, so the server considers the indicator variable $I_{4}=0$. $c_{2}$ stops the local update from being performed when the deadline approaches because it is a slow straggler and uploads a model that has not been updated at all with indicator variable $I_{2}=0$.

In step 6, straggler $c_{2}$ increases the $u_{2,t}$ value from 0 to 1. By setting $u_{2,t}$ as 1, the local dataset of $c_{2}$ is divided into two sub-datasets that are IID each other through SMS of CATA-Fed. $c_{2}$ reports the change of its data size to the size of the sub-dataset to the server. Meanwhile, some slow stragglers perform SMS multiple times, as $c_{5}$ and $c_{6}$ show in step 13 of Figure 9. For the case of $c_{6}$, despite the failure of local training, no more splits are made on the client. This is because further partitioning of its dataset may exceed the lower limit of the data size, π. Therefore, $u_{6,t+1}$ of $c_{6}$ does not increase from 1 anymore, whereas $u_{5,t+1}$ of $c_{5}$ increases from 1 to 2. In steps 7 and 14, the server selectively aggregates valid clients among the uploaded local update results and performs a global update.

Algorithm 2 is the pseudocode of CATA-Fed. The whole algorithm consists of a code for the server and a code for the clients. The global model training process of the central server is presented in lines 1 to 19 and the local update process of the participating client is described in lines 21 to 49. As a first step, the server clusters and groups entire connected clients via CFS and selects, $G_{x}^{*}$, a clients group with the highest priority (lines 3 to 11). From then, each participating client of the selected group performs local update (lines 12 to 15). In this process, the client performs adaptive local update as described in lines 26 to 36, and at the same time tracks the possibility of straggling (lines 45 to 49). If any client is determined to be a slow straggler, it performs SMS as represented in lines 38 to 41. After that, the server updates the global model via selective aggregation based on the upload results (line 16).

**Algorithm 2** Algorithm of CATA-Fed**Input:** Set of connected clients *C* , the number of global iteration *E* , deadline *T*, the number of cluster *P*, learning rate η, the number of participating clients *N*, lower limit size of sub-dataset π**Output:** Global model *W*
1:  **procedure**
Server(E,k)2:    Initialize $W_{0},A_{i}\left[0\right]\left(1\leq i\leq |C|\right)$3:    $X_{m}\left(1\leq m\leq P\right)\leftarrow$ client clustering according to Equation (Equation 20)4:    **for** global iteration 0≤t≤E−1 **do**5:     **if**
0<t , $S_{t-1}^{D}\neq \emptyset$ and detect report of client $c_{i}\in S_{t-1}^{S}$ **then**6:      Reclustering $X_{m}\left(1\leq m\leq P\right)$ according to Equation (Equation 20)7:     **end if**8:     Update $A_{i}\left[t\right]$ of client $c_{i}\in C$ according to Equation (Equation 21)9:     Grouping clients $G_{x}$ according to Equation (Equation 23)10:   Calculate priority $p_{x}$ according to Equation (Equation 24)11:   $S_{t}\leftarrow G_{x}^{*}$ according to Equation (Equation 25)12:   **for** each client $c_{i}\in S_{t}$ in parallel **do**13:    Broadcast $W_{t},T$ to client $c_{i}$14:    $W_{t}^{i},I_{i}\leftarrow$ ClientUpdate $\left(c_{i},W_{t},T\right)$15:   **end for**16:   Update global model $W_{t+1}\leftarrow \sum_{i=1}^{|S_{t}|}\psi_{i}I_{i}W_{t}^{i}$17:   $C\leftarrow C-S_{t}^{D}$18:  **end for**19:
**end procedure**
20:   21:**procedure**clientupdate($c_{i},W_{t},T$)22:  Initialize $I_{i}=1$23:  $n_{i}\leftarrow n_{i}+1$24:  Select training data $D_{i,j}$, $j=n_{i}\;mod\;\left(2^{u_{i,t}}\right)+1$25:  Replace local model $w_{t}^{i}$←$W_{t}$26:  Check start training time $\tau_{start}$27:  Working BreakProcess($T,\tau_{start},I_{i}$) in parallel28:  **for** local epoch 0≤e **do**29:   **if** $\left(T-\varepsilon_{i}\right)-\sum_{u=0}^{e-1}\tau_{u}^{i}<\tau_{mean}^{i}$ **then**30:    Break training31:   **else**32:    $w_{t,e+1}^{i}=w_{t,e}^{i}-\eta ▿F_{i}\left(w_{t,e}^{i},D_{i,j}\right)\times I_{i}$33:    Get training time $\tau_{e}^{i}$34:    Update average of training time $\tau_{mean}^{i}$35:   **end if**36:  **end for**37:  Upload $W_{t}^{i},I_{x}$ to server38:  **if**$I_{x}$ is 0 **then**39:   Update $u_{i,t}$ according to Equation (Equation 10)40:   Partitioning sub-dataset $D_{i,j},j\in \left[1,2^{u_{i,t}}\right]$ according to Equation (Equation 11)41:   Report $|D_{i,j}|,j\in \left[1,2^{u_{i,t}}\right]$ to server42:  **end if**43:
**end procedure**
44:   45:**procedure**breakprocess($T,\tau_{start},I_{i}$)46:  **if** $\left(T-\varepsilon_{i}\right)\leq \tau_{now}-\tau_{start}$ **then**47:   Break training & $I_{i}=0$48:  **end if**49:
**end procedure**



## 5. Simulation Results

In this section, to evaluate the performance of CATA-Fed, extensive simulations are conducted as follows: (1) performance of ALU with SMS, (2) performance of CFS with PF scheduler, (3) performance of CATA-Fed under statistical heterogeneity conditions, and (4) performance of CATA-Fed under long-tail distribution. The performance of CATA-Fed is compared with three FL schemes (FedAVG [4], FedProx [18], TiFL [19]). There are two main performance metrics in these simulations. One is accuracy, which means the inference hit ratio for 10,000 test data sheets of the benchmark dataset. The other is the training speed, which means the number of global iterations (communication rounds) to reach a target accuracy. In the simulation, the target accuracy is defined as a value 5% lower than the peak accuracy and is expressed as a horizontal line in the simulation result graphs.

### 5.1. Simulation Setups

In the simulations, a total of 4000 clients are connected to the network and the central server selects 1% of them as the participants in each global iteration for training the global model. All benchmark data (MNIST, Fashion-MNIST, CIFAR-10) contain 10 classes, and each class consists of 5000 pieces of training data and 1000 pieces of test data. To distribute the benchmark dataset with the limited size into large-scaled network (4000 connected clients), image augmentation is performed on the training dataset to solve the data duplication problem [41]. The detailed tuning of the augmentation is as follows: image rotation range = [−15, 15] degree, image horizontal flip = 50% of probability, image width shift range = 10% of the original image, image height shift range = 10% of the original image. We also normalized the value of every element in the data to [0,1]. The global model has a CNN layer ([32 × 32], [32 × 32], [64 × 64], [64 × 64], [128 × 128], [128 × 128]) with a kernel of [3 × 3] of 6 layers and 3 dense layers (1024,512,256). After the convolution layer, a model with a Maxpooling layer of [2 × 2] and a DropOut layer of rate 0.2 is constructed. ReLU is used as the activation function, and an output layer with Softmax is used. The optimizer is SGD, the learning rate is 0.01 and the batch size is 32. The data size of each client is randomly determined between 100 and 3000. We also assume the time cost for uploading $\varepsilon_{i}=0$. The minimum training data size π=100.

In the simulation, basically, each participating client in the target comparison scheme is set to conduct fixed numbers (K=5) of local updates in a given global iteration. Exceptionally, the client in FedProx performs a maximum of *K* local updates as long as the deadline is not exceeded. The unit value of the deadline *T*, (T=1), is set by the average time for all the connected clients to perform the five numbers of local updates under the ideal conditions in FedAVG (without any failure of uploading the results of local updates). The deadline value of *T* calculated from the above is also referred in the simulations of the other schemes.

### 5.2. Performance of ALU with SMS

#### 5.2.1. Impact of Deadline

In this section, the simulation results are presented in Figure 10 that show the performance of ALU in CATA-Fed according to the deadline time T=1 and T=0.5. Every connected client has IID local data with all classes, and each class has the same amount of data. Forty clients are randomly selected from the entire connected clients to perform local training in each global iteration. In addition, no disconnected client is assumed, and the computing power of each client is assumed to be the same. In the case of FedAVG (T= inf), the clients perform training without a deadline so that all the participating clients successfully upload the local update result without a disconnection. Meanwhile, the two 3-tuple of schemes [CATA-Fed (T=1), FedAVG (T=1), and FedProx (T=1)] and [CATA-Fed (T=0.5), FedAVG (T=0.5) and FedProx (T=0.5)] have a deadline time of the same length in each global iteration, respectively.

In the simulation with [MNIST, Fashion-MNIST, and CIFAR-10], CATA-Fed (T=1) achieves [2.0×, 1.64×, and 1.55×], [2.27×, 2.28×, and 2.42×], and [2.63×, 1.65×, and 2.13×] faster training speed than FedAVG (T= inf), FedAVG (T=1), and FedProx (T=1), respectively. These training speed improvements of CATA-Fed are because the optimal point of the objective function of the participating clients can be approximated at a lowered communication cost through ALU, compared to fully aggregating schemes (without client dropout) such as FedAVG (T= inf) and FedProx. This acceleration of the local model convergence influences the number of global iteration to be reduced for global model convergence. In addition, FedAVG (T= inf) has a higher training speed than FedAVG (T=1). This is because FedAVG (T= inf) does not experience a drop in clients while FedAVG (T=1) may experience a drop in some big clients, owing to the deadline where the drop slows down global convergence, wasting computing resources. (The simulation results are the testing accuracy according to the number of global iterations, not to the real time. This may also make a difference.) On the other hand, CATA-Fed (T=1) experiences much less dropping of clients, so that can reduce the waste of resources.

Meanwhile, in the figure, CATA-Fed (T=1) achieves similar or slightly higher test accuracy than FedAVG (T= inf), FedAVG (T=1) and FedProx (T=1). In MNIST, CATA-Fed (T=1) achieves 0.65%, 0.92% and 0.97% higher peak accuracy value than FedAVG (T= inf), FedAVG (T=1), FedProx (T=1) during 100 rounds of global iteration, respectively. In Fashion-MNIST, CATA-Fed (T=1) has a higher peak accuracy than FedAVG (T= inf), FedAVG (T=1) and FedProx (T=1) by 0.94%, 1.42% and 1.98% during 300 rounds as shown in Table 2. In CIFAR-10, CATA-Fed (T=1) attains 0.53%, 4.05% and 2.82% higher peak accuracy than FedAVG (T= inf), FedAVG (T=1) and FedProx (T=1) during 1000 rounds, respectively.

More simulations are conducted for the cases of shortened deadline time to T=0.5. In the case of CIFAR-10, the training speed of FedAVG (T=0.5) and FedProx (T=0.5) are reduced to 0.61× and 0.71× of T=1. On the other hand, the training speed of CATA-Fed (T=0.5) is decreased to 0.72× of T=1. This shows that CATA-Fed is more robust than FedAVG and FedProx for the short local training intervals. More notably, CATA-Fed (T=0.5) outperforms FedAVG (T= inf) with all the benchmark datasets in terms of training speed. It can be inferred that the mitigation for the straggling client that hinders global model convergence is being effectively performed by SMS as well as ALU under the setting of a short deadline time.

#### 5.2.2. Impact of Straggler Ratio

In this section, extensive simulations are conducted to evaluate the robustness of ALU of CATA-Fed varying, λ, the ratio of tentative stragglers among the connected client. The tentative straggler is defined as a client with half the computing power of the normal client. In the simulations, it is assumed that all clients have IID local data and there is no disconnected client.

As shown in Figure 11, the peak accuracies of FedAVG and FedProx decrease by [3.39% and 3.74%] and [0.72% and 2.91%] in Fashion-MNIST and CIFAR-10 as λ increases from 0 to 20. On the other hand, the peak accuracy of CATA-Fed decreases only by 0.25% and 0.8% as λ increases from 0 to 20 in Fashion-MNIST and CIFAR-10, respectively. In addition to this, the simulation results show that the performance degradation in terms of the training speed of CATA-Fed is much smaller than that of FedAVG and FedProx. Therefore, it can be inferred that CATA-Fed is more robust than FedAVG and FedProx in global model training over heterogeneous systems. The first reason for the robustness of CATA-Fed is that ALU reduces the client drop probability when a straggling client is selected. The second reason is that the probability of selecting straggling clients is reduced as the more global iteration is repeated because SMS manages to reduce the number of straggling clients when it finds the straggling clients.

### 5.3. Performance of CFS with PF Scheduler

#### 5.3.1. Impact of Cluster

As shown in Figure 12, the performance results of CFS are evaluated, varying the number of clusters under the condition of statistical heterogeneity. To this end, in the simulations, the clients have a biased data distribution (non-IID data) with parameter H=1. For each client, 90% of the local data is designed to consist of *H* randomly chosen classes out of 10. The remaining (10−H) classes are randomly distributed on the remaining 10% (hereinafter called class bias). ALU, SMS, and PF scheduler are implemented in the clients of CATA-Fed. *P* is the number of clusters, and no clustering is applied in CATA-Fed (P=1). In the case of TiFL (uniform mode in [19]), entire connected clients are categorized into *P* tiers (clusters) according to the size of the local dataset. To perform a global iteration, tiers are chosen with uniform probability in advance, and then participating clients are randomly selected within the selected tier. In addition, no disconnected client is assumed, and the computing power of each client is assumed to be the same.

Since training on non-IID data may cause variations in test accuracy, the average values of the observed accuracy values during the last 50 rounds of global iteration are presented in Table 3. CATA-Fed (P=8) achieves 3.28% and 7.75% higher average accuracy than CATA-Fed (P=1) in Fashion-MNIST and CIFAR-10, respectively. From these comparison results, the positive effect of clustering can be confirmed in terms of test accuracy. By using clustering, in every global iteration, CATA-Fed balances the magnitude of the gradients across the participating clients with the biased local data. This more fair global update results in the improvement of the model accuracy. Meanwhile, Table 3 shows that the test accuracy gradually improves as the number of cluster increases. However, the performance improvement seems to be saturated as the number of clusters increases. It is not a trivial task to find the optimal point of the cluster number which can vary depending on the number of clients, the composition of the dataset, and the degree of bias. Thus, we keep this as a future work that must be solved.

Meanwhile, Figure 12c shows that the variation widths in the test accuracy of clustered CATA-Fed (P=2,4,8) are reduced compared to those of non-clustered CATA-Fed (P=1) and FedAVG. The variance in the test accuracy values of each scheme is summarized in Table 3. In particular, when the simulation is conducted with CIFAR-10, variance of the accuracy for CATA-Fed (P=8) is 0.228 while FedAVG shows 10.876 of variance. This means that the clustering (CFS) can enhance the training stability of the federated learning under heterogeneous data distributions, which enables global updates to make more accurate global model.

Through the simulations, the effects of ALU and SMS can also be found under heterogeneous data conditions. CATA-Fed (P=1) achieves 6.12% and 6.43% higher average accuracy than FedAVG in Fashion-MNIST and CIFAR-10. Moreover, CATA-Fed (P=4) obtains 5.53% and 8.48% higher average accuracy than TiFL (P=4), respectively. According to the these comparison results, it is inferred that straggling clients management of ALU and SMS of CATA-Fed also affect the increase in accuracy by reducing the drop of clients with unique data. That is to say, ALU and SMS enhance the generalization performance of the federated learning.

#### 5.3.2. Impact of PF Scheduler

In this section, we evaluate the fairness of the training opportunity across clients and performance of PF scheduler of CATA-Fed. In this simulation, it is assumed that there is no disconnected client and 1 class is biased over non-IID setting (H=1). ALU, SMS, and the clustering are implemented in CATA-Fed. As a comparison scheme, CATA-Fed round robin (RR: P=4) generates four clusters and sequentially selects the clusters for the local training. When RR (P=4) selects a cluster, then it randomly selects participating clients within the selected cluster. CATA-Fed random selection (P=1) is a scheme without clusters that randomly selects the participating clients from all the clients, such as FedAVG.

In this experiment, Jain’s fairness index is introduced to evaluate the fairness of the PF scheduler of CFS. Raj Jain’s equation is widely used to measure the fairness in the quality of service of multiple clients in telecommunication engineering. In CATA-Fed, the result values of the fairness lie on the range from $\frac{1}{\left|C\right|}$ (worst case) to 1 (best case), and the best case can be achieved when all the clients receive the same number of the selection. Raj Jain’s equation is expressed as
(26)$$J\left(x_{1},x_{2},\ldots ,x_{\left|C\right|}\right)=\frac{{\left(\sum_{i=1}^{\left|C\right|}x_{i}\right)}^{2}}{\left|C\right|\times \sum_{i=1}^{\left|C\right|}x_{i}^{2}}$$
where |C| is the total clients connected in network, $x_{i}$ is the number of client $c_{i}\in C$ selections. In Figure 13a, in terms of the fairness index, the PF scheduler of CATA-Fed converges to 0.988 on average. It is slightly higher than 0.977 of random selection but with almost similar accuracy. It is higher than 0.878 of RR. It can be seen that learning can be done fairly with the PF scheduler, and the generalization performance may not be hindered from the PF scheduler. This is also confirmed through the simulation results in Figure 13b. The only difference between CATA-Fed (P=1) and CATA-Fed Random (P=1) in the figure is the selection method of the participating clients (PF scheduling and random selection). Both schemes achieve similar test accuracy, indicating that the generalization performance of the PF scheduler is not compromised.

The role of the PF scheduler in CATA-Fed is to provide fair learning opportunities to clients and at the same time determine the order of selecting the generated clusters. By PF, each cluster is selected with a rate proportional to its size. The performance comparison between selecting clusters in a dedicated order and selecting clusters through the PF scheduler can be seen by examining the accuracy of CATA-Fed PF (P=4) and CATA-Fed RR (P=4) in Figure 13b. In the figure, PF has a 3.39% higher average accuracy than RR and the gap of accuracy values between PF and RR are gradually widened. From this, it can be inferred that cluster selection through the PF scheduler outperforms the round-robin method.

### 5.4. Performance of CATA-Fed under Statistical Heterogeneity Conditions

In this section, the simulation is conducted to observe the impact of statistical heterogeneity on the performance of CATA-Fed varying parameter H∈[1,10] with CIFAR-10. If *H* is small, the clients have a more biased data distribution. And if H=10, the clients have IID local data.

As shown in Figure 14, the decrease in accuracy of CATA-Fed as the degree of class bias increases is much less than that of FedAVG and TiFL. More specific values can be confirmed through Table 4. As *H* goes from 10 to 1, the average accuracy degradation of TiFL is 10.17%, while that of CATA-Fed is only 6.58%. From the results, the generalization effect of straggler management through ALU and SMS on the global model can be confirmed once again and it can be inferred that CATA-Fed is more robust than TiFL under statistical heterogeneity conditions. Meanwhile, the variance of accuracy for FedAVG becomes 10.876 when H=1, while that of CATA-Fed and TiFL, cluster-based schemes, are still 0.188 and 0.529. This means that the clustering approach enhances the stability of FL with non-IID data.

### 5.5. Performance of CATA-Fed under Long-Tail Distribution

In this section, extensive simulation is conducted to evaluate the performance of CATA-Fed when the data distribution across clients is long-tail. In the simulation, the local data size of a client in normal CATA-Fed, FedAVG, and TiFL is randomly determined within [100, 3000]. On the other hand, the local data size across the clients of each scheme with long-tail distribution (CATA-Fed LT, FedAVG LT, and TiFL LT) are designed to be dispersed over a positive skewed distribution with a long tail to the right. To this end, 40%, 30%, and 20% of the clients have random data sizes within the ranges of [100, 300], [300, 500], and [500, 1000]. The remaining 10% of the clients have random data sizes in [1000, 3000].

Note that the differences in data sizes among the participating clients may become larger in the long-tail distribution. In this case, the weighted averaging process can be biased by the bigger clients in the global model update. Figure 15 and Table 5 show the test accuracy of CATA-Fed, FedAVG and TiFL under the designed long-tail distribution. CATA-Fed LT achieves 99.9%, 96.7%, and 95.76% of normal CATA-Fed accuracy with MNIST, Fashion-MNIST, and CIFAR-10, respectively. In the same manner, FedAVG LT and TiFL LT achieves [98.7%, 89.4%, 85.32%] and [98.6%, 94.4%, 90.92%] of normal FedAVG and TiFL accuracy with [MNIST, Fashion-MNIST, and CIFAR-10], respectively. From these results, it can be inferred that CATA-Fed is more robust than FedAVG and TiFL to the skewed distribution. This is because CFS in CATA-Fed can alleviate the bias of the relatively large clients in weighted averaging process by utilizing clustering. As shown in Table 5, CATA-Fed outperforms TiFL in terms of accuracy variance. CFS of CATA-Fed considers outliers and generates the moderate ranges for the clusters by using IQR, while TiFL evenly divides the entire distribution range without considering outliers. Therefore, intervals of the clusters in CATA-Fed are relatively smaller than those in TiFL, which provides a more fair global model update.

## 6. Conclusions and Future Work

In this paper, we proposed a cluster-driven adaptive training approach for federated learning (CATA-Fed) to address the straggler problem and training performance degradation under statistical heterogeneous conditions. CATA-Fed alleviates the influence of the straggling clients by applying adaptive local update and the straggler mitigation scheme. In addition, CATA-Fed reduces the divergence probability of the global model under training with non-IID datasets by utilizing client clustering and proportional, fair client selection. The results of the extensive simulations on three realistic FL benchmark datasets confirm that CATA-Fed outperforms the comparison schemes in terms of training speed, test accuracy, and robustness of FL under the diverse practical environments. As a future work, the optimization issues of the parameters in CATA-Fed to maximize FL performance will be discussed under diverse training scenarios.

## Figures and Tables

**Figure 1 sensors-22-07061-f001:**
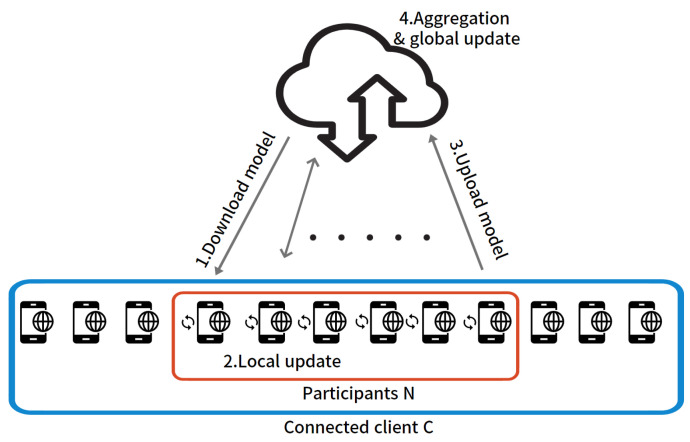
FL system architecture.

**Figure 2 sensors-22-07061-f002:**
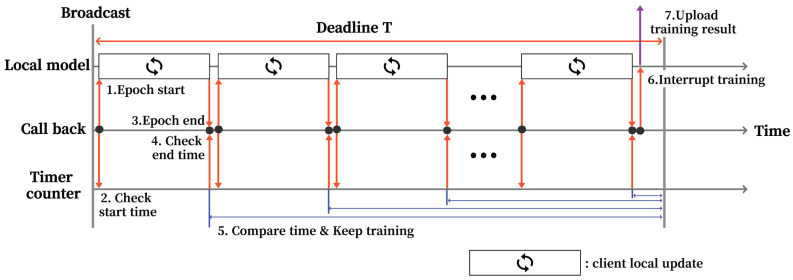
Adaptive local update of the participate client.

**Figure 3 sensors-22-07061-f003:**
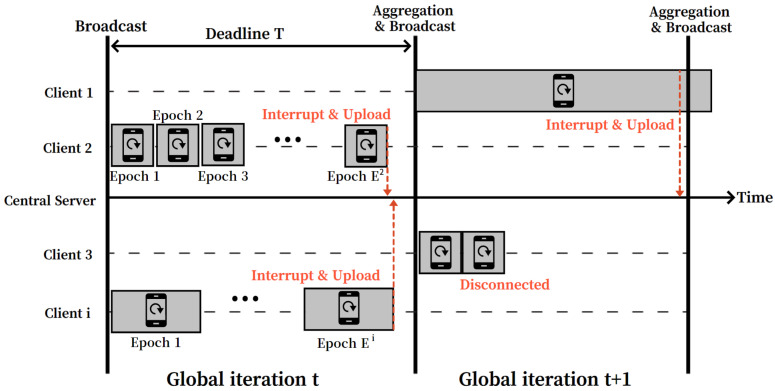
Examples of the diverse client training cases.

**Figure 4 sensors-22-07061-f004:**
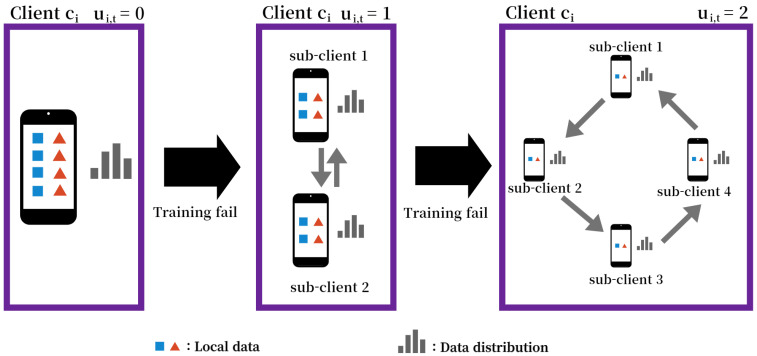
Example of SMS.

**Figure 5 sensors-22-07061-f005:**
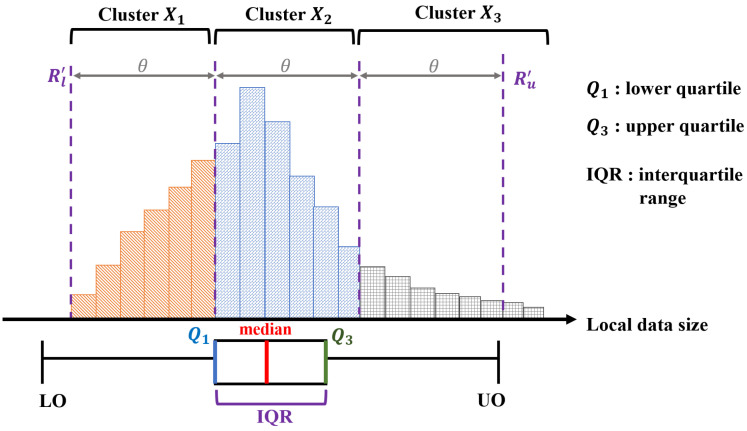
Example of the clients clustering when P=3.

**Figure 6 sensors-22-07061-f006:**
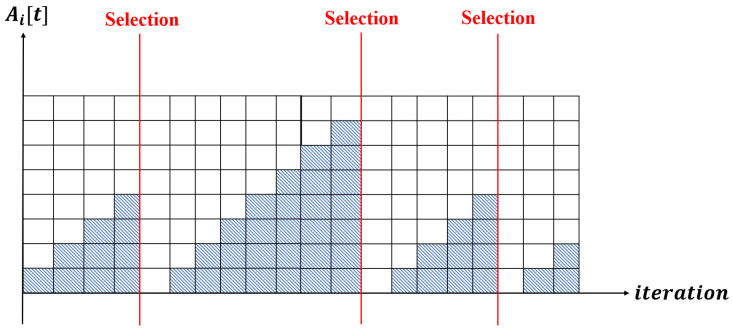
Example of $A_{i}\left[t\right]$ of client $c_{i}$.

**Figure 7 sensors-22-07061-f007:**
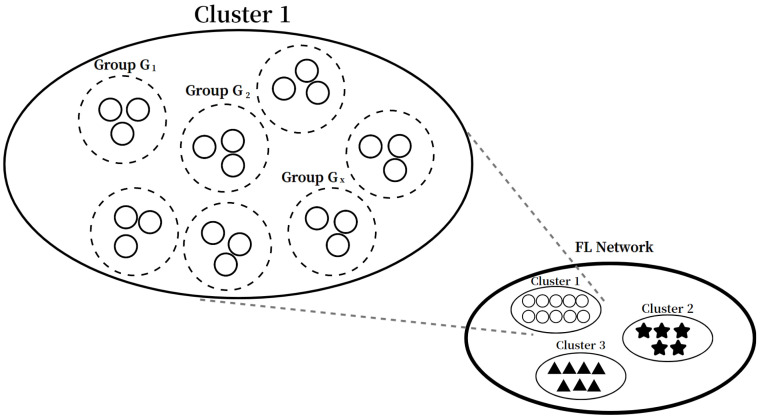
Example of the grouping clients in clusters when $|s_{t}|=3$.

**Figure 8 sensors-22-07061-f008:**
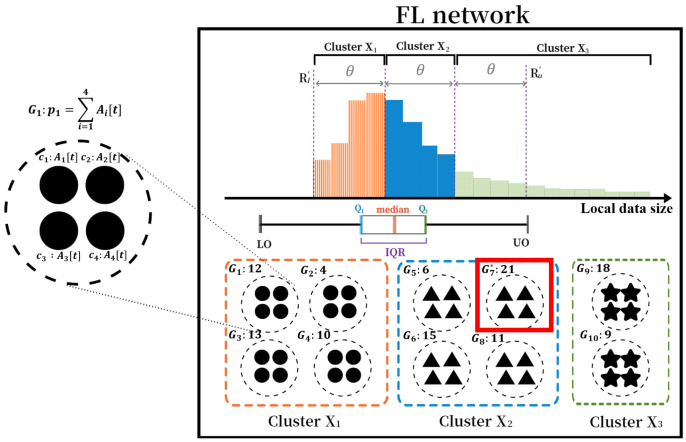
Example of CFS.

**Figure 9 sensors-22-07061-f009:**
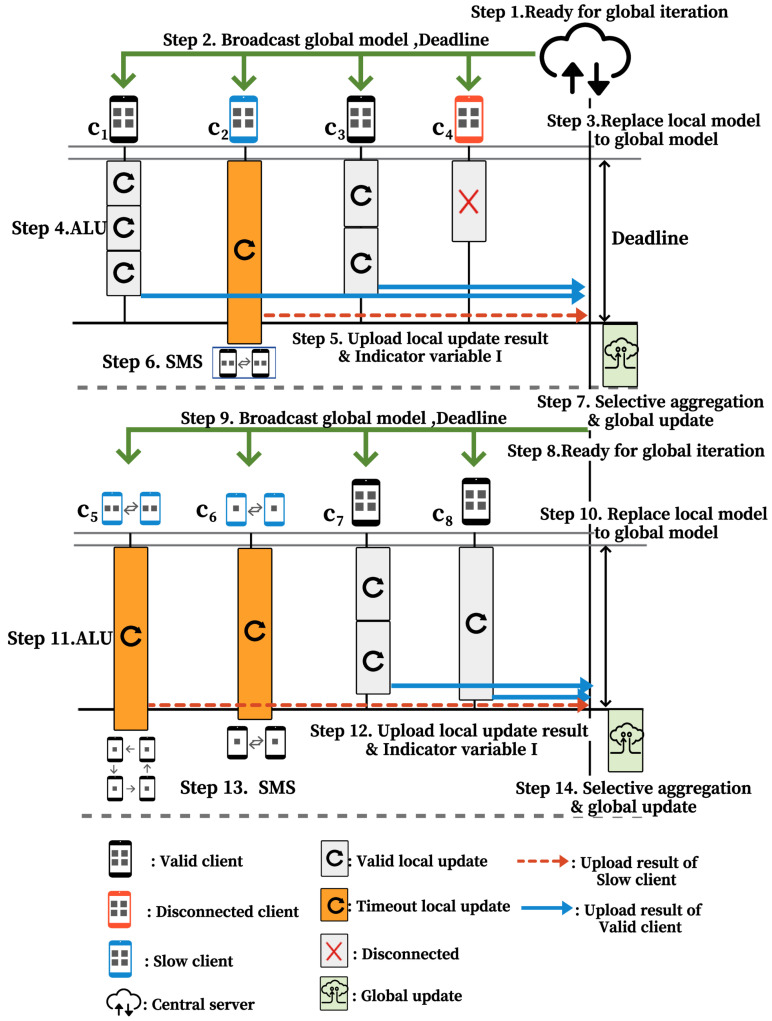
Operational example of CATA-Fed.

**Figure 10 sensors-22-07061-f010:**
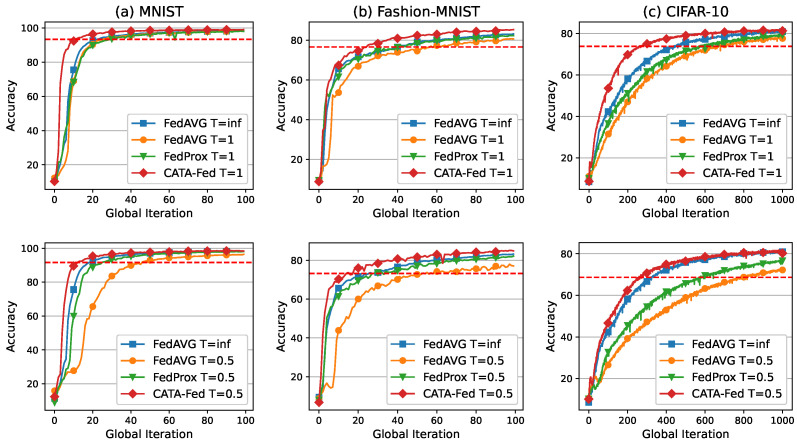
Test set accuracy vs. global iterations for (**a**) MNIST, (**b**) Fashion-MNIST, and (**c**) CIFAR-10 under IID setting with deadline T=1 (**top**) and T=0.5 (**bottom**).

**Figure 11 sensors-22-07061-f011:**
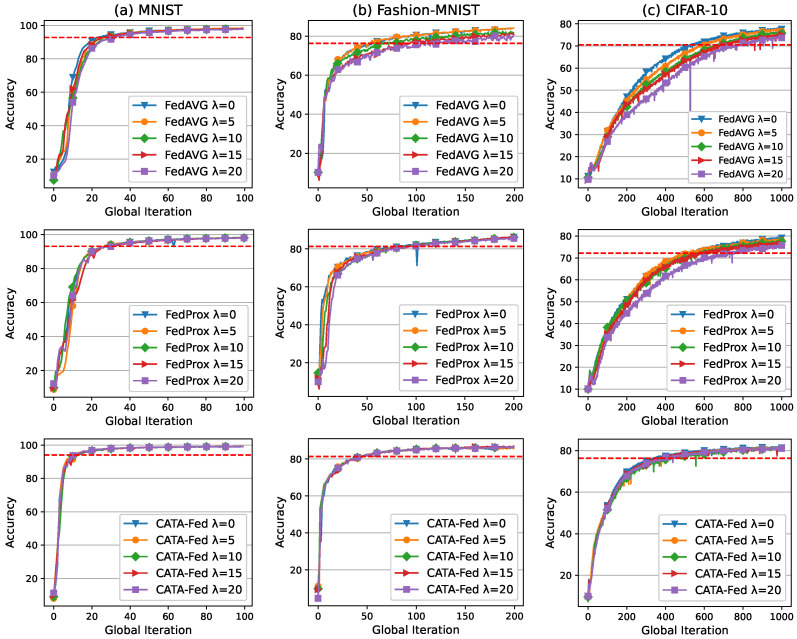
Test set accuracy of FedAVG (**top**), FedProx (**middle**) and CATA-Fed (**bottom**) vs. global iterations for (**a**) MNIST, (**b**) Fashion-MNIST, and (**c**) CIFAR-10 under IID setting with tentative straggler ratio λ=[0,5,10,15,20], deadline T=1.

**Figure 12 sensors-22-07061-f012:**
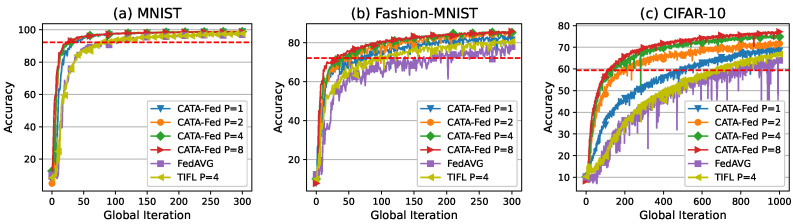
Test set accuracy vs. global iterations for (**a**) MNIST, (**b**) Fashion-MNIST, and (**c**) CIFAR-10 under the condition of in non-IID setting with the number of cluster P=[1,2,4,8], deadline T=1, the number of biased class H=1.

**Figure 13 sensors-22-07061-f013:**
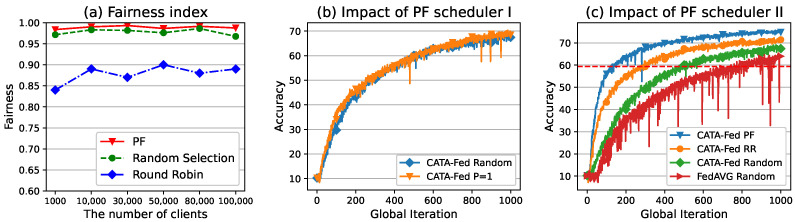
(**a**) The number of clients vs. fairness index of Raj Jain, (**b**) test set accuracy of CATA-Fed PF (P=1) vs. global iterations for CIFAR-10, and (**c**) test set accuracy of CATA-Fed PF (P=4) vs. global iterations for CIFAR-10 with deadline T=1, the number of biased class H=1.

**Figure 14 sensors-22-07061-f014:**
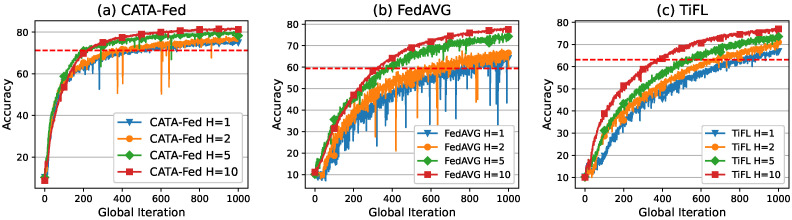
(**a**) Test set accuracy of CATA-Fed vs. global iterations, (**b**) test set accuracy of FedAVG vs. global iterations and (**c**) test set accuracy of TiFL vs. global iterations for CIFAR-10 under the condition of non-IID setting with class bias H=[1,2,5,10], the number of cluster P=4, and deadline T=1.

**Figure 15 sensors-22-07061-f015:**
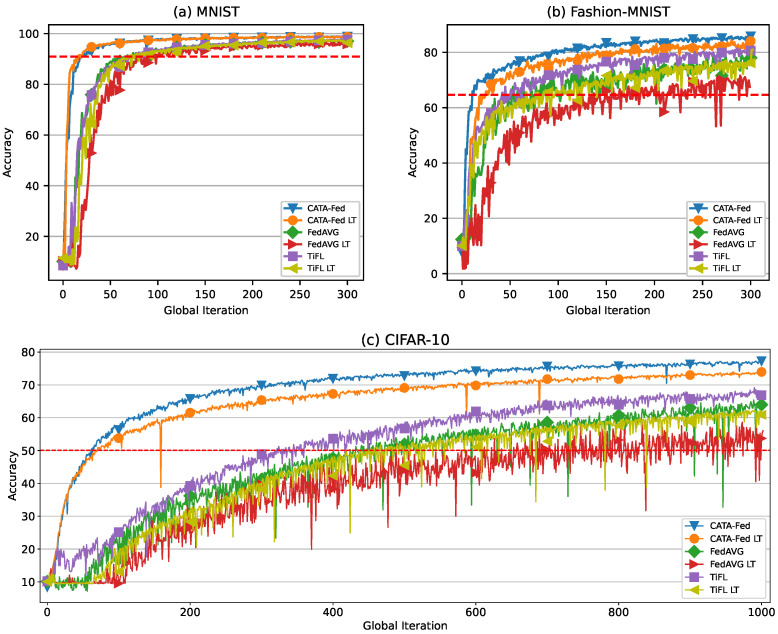
Test set accuracy vs. global iterations for (**a**) MNIST, (**b**) Fashion-MNIST, and (**c**) CIFAR-10 under the condition of long-tail distribution with the number of cluster P=8, deadline T=1, and class bias H=1.

**Table 1 sensors-22-07061-t001:** Features of FL schemes.

Topic	Target Heterogeneity Issue	Client Grouping	Client Selection	Local Model Aggregation	No. LU ^1^
CATA-Fed	System + Statistical	Cluster	Scheduling	Synchronous	Flexible
FLANP [30]	System	-	Sequential	Synchronous	Fixed
FedProx [18]	System	-	Random	Synchronous	Flexible
FedAT [33]	System	Tier	Random	Synchronous + Asynchronous	Fixed
SCAFFOLD [27]	Statistical	-	Random	Synchronous	Fixed
FAVOR [28]	Statistical	-	DRL Agent	Synchronous	Fixed
CFL [29]	Statistical	Cluster	Random	Synchronous	Fixed
FedBuff [35]	System + Statistical	-	Random	Asynchronous	Fixed
TiFL [19]	System + Statistical	Tier	Scheduling	Synchronous	Fixed
Oort [36]	System + Statistical	-	Adaptive	Synchronous	Fixed
FedAsync [37]	System + Statistical	-	Random	Asynchronous	Fixed

^1^ The number of local updates in a given global iteration.

**Table 2 sensors-22-07061-t002:** Peak test accuracy in Fashion-MNIST with global iterations (round).

Scheme	100 Rounds	200 Rounds	300 Rounds
CATA-Fed T= 1	85.33%	88.80%	89.75%
CATA-Fed T= 0.5	84.88%	87.88%	90.24%
FedAVG T= inf	83.21%	86.98%	88.81%
FedAVG T= 1	80.65%	86.59%	88.33%
FedAVG T= 0.5	77.10%	83.78%	86.61%
FedProx T= 1	82.24%	86.79%	87.77%
FedProx T= 0.5	82.06%	86.33%	87.18%

**Table 3 sensors-22-07061-t003:** Mean and variance of accuracy values of last 50 rounds of global iterations.

Scheme	Fashion-MNIST		CIFAR-10	
	Mean	Variance	Mean	Variance
CATA-Fed P= 8	85.34%	0.251	76.68%	0.228
CATA-Fed P= 4	85.09%	0.266	74.96%	0.188
CATA-Fed P= 2	84.52%	0.357	71.59%	0.529
CATA-Fed P= 1	82.06%	2.216	68.93%	0.721
FedAVG	75.94%	3.113	62.50%	10.876
TiFL P= 4	79.56%	1.157	66.48%	0.596

**Table 4 sensors-22-07061-t004:** Comparison of mean, variance of accuracy in CIFAR-10 according to the degree of class bias.

Class Bias	CATA-Fed		FedAVG		TiFL	
	Mean	Variance	Mean	Variance	Mean	Variance
H=10	81.54%	0.029	77.62%	0.326	76.65%	0.157
H=5	79.45%	0.149	73.81%	1.022	72.91%	0.749
H=2	76.52%	0.216	65.33%	2.608	69.58%	0.659
H=1	74.96%	0.188	62.50%	10.876	66.48%	0.529

**Table 5 sensors-22-07061-t005:** Comparison of average test accuracy according under the long-tail distribution.

Class Bias	MNIST		Fashion-MNIST		CIFAR-10	
	Mean	Variance	Mean	Variance	Mean	Variance
CATA-Fed	98.72%	0.009	85.02%	0.311	76.68%	0.082
CATA-Fed LT	98.63%	0.013	82.24%	1.015	73.43%	0.128
FedAVG	96.98%	0.021	75.94%	3.113	62.50%	10.876
FedAVG LT	95.74%	0.072	67.83%	17.147	53.33%	14.736
TiFL	97.33%	0.075	79.56%	1.157	67.13%	0.718
TiFL LT	95.99%	0.129	75.15%	3.988	61.04%	1.748

## Data Availability

The results presented in this study are provided on a limited basis upon request of the corresponding author. Data may not be made publicly available for reasons of data protection by the relevant organizations.

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
