# Peer review of "A Cluster-Driven Adaptive Training Approach for Federated Learning"

_sensors, 2022, doi:10.3390/s22187061_

Round 1
Reviewer 1 Report
This paper proposes a cluster-driven adaptive training approach called CATA-Fed. CATA-Fed consists of two stages to address the straggler issue and non-IID issue respectively. The first stage utilizes 1) adaptive local update (ALU) to allow a client to adapt the maximum number of its local updates internally given a deadline; 2) straggler mitigating scheme (SMS) to split the data of stragglers so that local training can be completed within the interval of a global iteration rather than dropping them directly. The second stage proposes a cluster-driven fair client selection scheme (CFS) to select clients with similar data sizes and perform a weighted averaging process. CATA-Fed is evaluated on three datasets “MNIST”, “FMNIST” and “CIFAR-10”, and the results show that the proposed mechanism outperforms FedAVG in terms of accuracy and training speed.
In general, the paper introduces the methodologies clearly, however, the paper is mediocre. The experiments need to be enhanced (e.g., adding baselines) and the presentations need to be improved (e.g., figures, related work, the comparison with related methods, and the algorithm of CATA-Fed). The details of the comments are as follows.
1) The English should be improved, since a lot of grammar errors need to be corrected in this paper, such as
l Line 41: clients client, environments environment;
l Line 65-66: the updated gradients which is computed locally at the edge aggregated by the central server the updated gradients computed locally at the edge are aggregated by the central server;
l Line 77: does do;
l Line 88: determines determine, deadlines deadline;
l Line 77: and communication technologies communication technologies;
l Line 129-130: And this improve the bottleneck and reduce And this improves the bottleneck and reduces;
l Line 184: tth (all the tth) t-th;
l Line 207: complete completes;
l Line 209: in a third process in the third process;
l Line 212: in a fourh process in the fourth process;
l Line 217: distribute distributes;
l Line 317: the the the;
l Line 366-367: satisfies following conditions satisfies the following conditions.
2) The writing style is long-winded and many sentences/ paragraphs are redundant, e.g.,
l Line 300-302: During this time, participating clients try to update the local model as much as possible by means of ALU. That is to say, each client performs as many local updates as possible during local training time T.
The writing should be succinct and the length of the paper needs to be reduced.
3) Figures (such as Figures 5, 6, 10, 11, 12, 13, 14, and 15) need to be redesigned to use different bar/line patterns not only colors so that they can be seen clearly when printed on a black/white printer.
4) FedAVG is the simplest FL baseline and there are many state-of-the-art methods. Thus, these more advanced baselines should also be included in the experiments to verify the proposed mechanism, such as
l FedProx (Federated Optimization in Heterogeneous Networks);
l FedAsync (Asynchronous Federated Optimization);
l Oort (Oort: Efficient Federated Learning via Guided Participant Selection).
5) There are many papers that also address the straggler and non-IID data issues by cluster, such as
l TiFL (TiFL: A tier-based federated learning system);
l FedAT (A Communication-Efficient Federated Learning Method With Asynchronous Tires Under Non-IID Data);
l CFL (Clustered federated learning: Model agnostic distributed multitask optimization under privacy constraints).
It is suggested to include some related cluster-based FL methods in the related work and compare them with the proposed mechanism.
6) Related work is weak without fine organization and comparison, and it is suggested to design a summarized table for readability.
7) The corresponding descriptions of Algorithms 1 and 2 are needed in sections 3 and 4 respectively for readability.
8) Why the target accuracy for MNIST, Fashion-MNIST, and CIFAR-10 is 80%, 70%, and 60% respectively in Figure 11? I have similar questions for Figures 12, 13, and 14. It seems that the final accuracy in the three datasets outperforms the corresponding target accuracy greatly. Moreover, the accuracy in CIFAR-10 is still climbing at the end of learning, and it is suggested to implement more iterations until the model converges.
Reviewer 2 Report
The manuscript has proposed a cluster-driven adaptive training approach for federated learning to address straggler problem and training performance under statistical heterogeneous conditions. It is an interesting case study and the authors have documented it well. The technical contribution is significant and the results are promising. I would like to recommend the article for publication in its present format.
Reviewer 3 Report
In this paper, the authors proposed a cluster-driven adaptive training approach (CATA-Fed) to enhance the performance of FL training in practical environment, which can handle non-IID data and stragglers. Compared with the previous FL scheme (FedAVG), the proposed scheme can achieve faster training speed and higher accuracy in non-IID setting. Followings are my suggestions on this paper:
1. The system model part of the paper is only a brief introduction to the federated learning process, and it lacks a summary and sorting out of the main problems faced by the model in practice. Although some clarifications and starting points are given in the introduction, it is natural to present the problem in the system model and then introduce the proposed algorithm.
2. In the proposed adaptive local update (ALU) method, the participating client actively determine the maximum number of local updates comparing the remaining time to deadline and the average epoch time for its local update. However, how to ensure that the client is fully trained and converged under the maximum number of local updates?
3. In Figure 9, the authors present an example of CATA-Fed. In the first global iteration, in my opinion, c2 does not contribute to the global model update based on the SMS scheme. Only when c2 is selected next time, the model parameters of c2 will be uploaded to the server to participate in the update. Therefore, it should not be represented by blue arrows like c1 and c3.
4. In the simulation part, there is only one comparison algorithm. There are already some works to solve the non-IID problem, it would be more convincing to compare with more algorithms.
5. There are some typos in this paper, for example, the “A new straggler” in line 89 should be “a new straggler”.
6. The full name is required when referring to an abbreviation for the first time (e.g., SGD in line 127).
Round 2
Reviewer 1 Report
Thanks for addressing my concerns. The authors have added new experiments with suggested baselines and improved the presentation of the paper. It seems to clear up the ambiguities. I have three more suggestions:
1) Table I should include the proposed mechanism as well for convenient comparison.
2) The font in the figures (e.g., Figure 1-9) needs to be enlarged to let readers see clearly.
3) Adding citations to well-known worldwide journals (especially publications in recent years) would inspire people in its research community to take an interest in this presentation. For example, the following papers might be cited in your work.
l You L, Liu S, Chang Y, et al. A triple-step asynchronous federated learning mechanism for client activation, interaction optimization, and aggregation enhancement[J]. IEEE Internet of Things Journal, 2022.
l Nguyen D C, Ding M, Pathirana P N, et al. Federated learning for internet of things: A comprehensive survey[J]. IEEE Communications Surveys & Tutorials, 2021, 23(3): 1622-1658.
l Feng C, Yang H H, Hu D, et al. Mobility-Aware Cluster Federated Learning in Hierarchical Wireless Networks[J]. IEEE Transactions on Wireless Communications, 2022.
Reviewer 3 Report
There are still some typos, especially in the related work part. It's suggested to go through the whole paper and check carefully.
1. In line 135, the "Various" should be "various".
2. When introducing related work, the author's singular and plural should be consistent. Specifically, "Reisizadeh et al. propose" in line 136 and "Chen et al. proposes" are inconsistent. In addition, the past tense is more appropriate here from my point of view.
